# Phenolic Compounds in Berries of Winter-Resistant *Actinidia arguta* Miq. and *Actinidia kolomikta* Maxim.: Evidence of Antioxidative Activity

**DOI:** 10.3390/antiox13030372

**Published:** 2024-03-19

**Authors:** Laima Česonienė, Viktorija Januškevičė, Sandra Saunoriūtė, Mindaugas Liaudanskas, Vaidotas Žvikas, Ričardas Krikštolaitis, Pranas Viškelis, Dalia Urbonavičienė, Paulina Martusevičė, Marcin Zych, Remigijus Daubaras, Aistė Balčiūnaitienė, Jonas Viškelis

**Affiliations:** 1Research Institute of Natural and Technological Sciences, Vytautas Magnus University, LT-40444 Kaunas, Lithuania; sandra.saunoriute@vdu.lt (S.S.); dalia.urbonaviciene@lammc.lt (D.U.); aiste.balciunaitiene@vdu.lt (A.B.); 2Botanical Garden, Vytautas Magnus University, LT-46324 Kaunas, Lithuania; viktorija.januskevice@vdu.lt (V.J.); pranas.viskelis@lammc.lt (P.V.); paulina.martusevice@vdu.lt (P.M.); remigijus.daubaras@vdu.lt (R.D.); jonas.viskelis@lammc.lt (J.V.); 3Institute of Pharmaceutical Technologies, Lithuanian University of Health Sciences, LT-50166 Kaunas, Lithuania; mindaugas.liaudanskas@lsmu.lt (M.L.); vaidotas.zvikas@lsmu.lt (V.Ž.); 4Department of Mathematics and Statistics, Vytautas Magnus University, Studentų 10, Kaunas District, LT-53361 Akademija, Lithuania; ricardas.krikstolaitis@vdu.lt; 5Lithuanian Research Centre for Agriculture and Forestry, Institute of Horticulture, LT-54333 Kaunas, Lithuania; 6Botanical Garden, University of Warsaw, Aleje Ujazdowskie 4, 00-478 Warsaw, Poland; marcin.zych@uw.edu.pl

**Keywords:** cultivars, procyanidins, hydroxycinnamic acid, flavan-3-nols, flavones

## Abstract

Variations between fruit cultivars can significantly impact their biochemical composition. The present research examined the variability in the qualitative and quantitative content of phenolic compounds in berry extracts of *Actinidia kolomikta* and *Actinidia arguta* cultivars. Additionally, antioxidant activities of berry extracts were evaluated. The total phenolic, flavonoid, proanthocyanidin contents and hydroxycinnamic acid derivatives were determined using the appropriate methodologies. The average amount of phenolic compounds in *A. kolomikta* berries (177.80 mg/g) was three times higher than that of *A. arguta* (54.45 mg/g). Our findings revealed that berries of *A. kolomikta* and *A. arguta* accumulated, on average, 1.58 RE/g DW (rutin equivalent/g dry weight) and 0.615 mg RE/g DW of total flavonoids, 1439.31 mg EE/g DW (epicatechin equivalent/g dry weight) and 439.97 mg EE/g DW of proanthocyanidins, and 23.51 mg CAE/g DW (chlorogenic acid equivalent/g dry weight) and 5.65 mg CAE/g DW of hydroxycinnamic acid derivatives, respectively. The cultivars of both species were characterized by higher antioxidant activity of total phenolic compounds determined using CUPRAC and FRAP methods compared to the ABTS^•+^ method. The variability in phenolic compounds’ qualitative and quantitative content in tested berry extracts was evaluated by applying ultra-high performance liquid chromatography (UHPLC) coupled to mass spectrometry in tandem with electrospray ionization. Significant intraspecific differences in the amounts of total phenolic compounds, total flavonoid compounds, proanthocyanidins, and hydroxycinnamic acid derivatives were determined among cultivars. Four phenolic acids, eight flavonols, two flavones, and five flavon-3-ols were identified in the berry extracts.

## 1. Introduction

Over the past few decades, there has been a significant increase in consumer interest in various fruits and their health benefits [1]. This can be attributed to a combination of factors, such as heightened health awareness, improved access to nutritional information and adherence to government dietary guidelines; year-round availability facilitated by transportation advancements, growing interest in functional foods that align with health and wellness trends; and evolving snacking habits that favor healthier options, as well as a broader societal shift towards plant-based and sustainable dietary choices [2,3].

Growing demand for healthy food products motivates the scientific community to expand the search and investigation of potential superfruits [4]. Furthermore, neglected and underutilized indigenous and introduced berry plants could be considered new sources of vitamins [5]. Berries of *Actinidia chinensis* var. *deliciosa* (A. Chev.) C.F. Liang & A.R. Ferguson., commonly known as the kiwifruit, are a good source of biologically active compounds [6]. Kiwifruit contains a complex network of primary and secondary metabolites, notable levels of dietary fiber, minerals, and different phytochemicals such as carotenoids and polyphenolic compounds, contributing to the rich pharmacological profile [7,8]. However, this species is not winter hardy, so the cultivars are cultivated only in mild subtropical areas. In colder regions, they require specialized care to survive winter temperatures below −10 °C.

The species *Actinidia arguta* Miq. and *Actinidia kolomikta* Maxim. are frost hardy. Hence, selected cultivars could be cultivated in countries where winters are more severe [9]. In recent years, *A. arguta* has become the second most popular species, and it has recently become commercially available [10]. Thus, recently, the biologically active compounds in the berries and leaves of this species have been more extensively studied by other authors [11,12]. *A. kolomikta* is not widely grown when considering the cultivation of other *Actinidia* Lindl. species. However, recent studies have confirmed high levels of ascorbic acid and other phytochemical compounds with antioxidant properties [13,14].

Hardy kiwifruit species produce berries with edible skin, which are rich in secondary metabolites [12,15]. These berries attract attention not only for their nutrients, such as vitamins, carbohydrates, and minerals but also due to the accumulation of health-promoting compounds such as anthocyanins, flavonols, flavan-3-ols, chlorophylls, *β*-carotene, etc. [11].

Phenolic compounds are major secondary metabolites that play an essential role in the nutritional and organoleptic properties of berries and their derived products. Polyphenol-rich berries, such as cranberry, blackberry, honeysuckle, and blueberry, reduce DNA oxidation damage and defend the human body against the damaging effects of free radicals [16]. Flavonoids, a group of compounds with variable phenolic structures, are also found in berries [17]. A variety of flavonoids have been identified in leaves, berries, and other parts of different *Actinidia* species, including 32 flavones, 71 flavonols, 3 isoflavones, 16 flavanones, and 13 anthocyanins [18]. The functional role of flavonoids in plant drought resistance and freezing tolerance has been proven to be significant in breeding new cultivars [19]. Flavonoids are also valued for their antifungal, antiviral, antibacterial activity, and other health-promoting properties [20]. Hydroxycinnamic acid derivatives were determined in fruits, vegetables, and fruit seeds and were also distinguished for their potent antioxidant and anti-inflammatory properties [21,22].

The berries’ biochemical composition depends on various biotic and abiotic factors such as cultivar properties, ripening stage, cultivation conditions, and harvest time. Therefore, this study focuses on the content of phenolic compounds in berries of two species, *A. arguta* and *A. kolomikta*. Selected cultivars were chosen to determine the variety of phenolics between these two species and among cultivars. The results of these studies should support the development of hardy kiwi cultivation and select the best genotypes as potential donors of valuable characteristics for the selection of new cultivars.

## 2. Materials and Methods

### 2.1. Plant Material

The research was accomplished on five cultivars of *A. kolomikta* and five cultivars and hybrids of *A. arguta* grown in the *Actinidia* spp. germplasm collection of Vytautas Magnus University Botanical Garden. Each cultivar is represented by 3–5 plants. The collection is located in the Kaunas region, central lowland district of Lithuania, 76 m above sea level, WGS84 coordinates 54.87055° N and 23.91621° E. The average annual temperature is +6.9 °C, the average annual precipitation is 700 mm, and the mean temperature is +18.0 °C in July and −3.8 °C in January. The growth period of different *Actinidia* cultivars continues from 175 to 186 days [23]. Characteristics of berries are presented in Figure 1 and Table 1.

Berries were stored at −80 °C until analysis. Before analysis, the berries were lyophilized at a pressure of 0.01 mbar and a condenser temperature of −85 °C using a Zirbus lyophilizer (Zirbus Technology GmbH, Bad Grund, Germany). Lyophilized berries were ground using the knife mill Grindomix GM 200 (Retsch GmbH, Haan, Germany), and samples were stored in tightly closed containers until investigation. The dehydration level of samples was determined with a hygrometer Precisa 310 M (Precisa, Dietikon, Switzerland). For each sample, the procedure was repeated three times, and averages of the drying estimates were calculated.

### 2.2. Chemicals

Ethanol 96 (*v*/*v*) was purchased from SC Vilniaus degtinė (Vilnius, Lithuania). ABTS^•+^ (2,2′-Azino-bis (3-ethylbenzothiazoline-6-sulfonic acid), acetonitrile, aluminum chloride hexahydrate, ammonium acetate, apigenin, caffeic acid, chlorogenic acid, (+)-catechin, copper (II) chloride dihydrate, (−)-epicatechin, ferulic acid, Folin–Ciocalteu reagent, gallic acid monohydrate, hyperoside, iron (III) chloride hexahydrate, isorhamnetin-3-*O*-rutinoside, isoquercitrin, kaempherol-3-*O*-glucoside, kaempferol-3-*O*-rutinoside, luteolin-4-*O*-glucoside, luteolin-7-rutinoside, neochlorogenic acid, phloridzin, potassium persulfate, procyanidin B1, procyanidin B2, procyanidin C1, rutin, sodium carbonate, trifluoroacetic acid, trolox ((±)-6-hydroxy-2,5,7,8-tetramethylchromano-2-carboxylic acid), quercetin were purchased from Sigma-Aldrich GmbH (Steinheim, Germany). Distilled water was produced using the Milli-Q^®^ 180 (Millipore, Bedford, MA, USA) water purification system.

### 2.3. Preparation of Berry Extracts

The ethanolic extracts were prepared using 2.5 g of lyophilized powder and 40 mL of 70% ethanol. The samples were extracted for 10 min at 80 kHz and 1017 Win in an ultrasonic bath (Elmasonic P, Singen, Germany). After extraction, the ethanolic extracts were centrifuged for 2 min at 8500 rpm, at room temperature, using a Heraeus Biofuge Stratos centrifuge (Heraeus Holding GmbH, Haan, Germany). The supernatants were poured from the residues, filtered, and placed in wide-mouthed bottles, which were kept in a refrigerator at 4 °C until analysis. The obtained ethanolic extracts were filtered through 0.22 µm pore-size membrane filters (Carl Roth GmbH, Karlsruhe, Germany).

### 2.4. Determination of Bioactive Compounds Profile

The total phenolic content (TPC) in the tested extracts was determined using the Folin–Ciocalteu method and expressed as gallic acid equivalent in dry weight (mg GAE/g DW) [24]. The total flavonoid content (TFC) in the tested extracts was determined using the described methodology [25]. The total proanthocyanidin content in the tested extracts was determined using a reaction with DMAC (4-Dimethylaminocinnamalaldehyde) reagent [26]. The total content of hydroxycinnamic acid derivatives in the tested extracts was determined using a reaction with Arnow reagent and expressed as chlorogenic acid equivalent [27].

### 2.5. Evaluation of Phenolic Compounds in Berry Samples Using the UHPLC-ESI-MS/MS Technique

The variability in the qualitative and quantitative content of phenolic compounds in tested samples was evaluated by applying ultra-high performance liquid chromatography (UHPLC) coupled to a mass spectrometer, using a technique described and validated in an article by Gonzalez-Burgos et al. [28]. The analysis of the qualitative and quantitative content of phenolic compounds in the samples of the tested extracts was carried out using a liquid chromatography system “Waters ACQUITY UPLC^®^ H–Class” (“Waters”, Milford, MA, USA) with a tandem quadrupole mass detector “Xevo TQD” (Waters, Milford, MA, USA). Sorting out of the compounds was performed using a “YMC Triart C18” (100 Å, 100 × 2.0 mm; particle size 1.9 μm) column (“YMC”, Kyoto, Japan) with a pre-column.

The mass spectrometry parameters for the analysis of phenolic compounds are presented in Table 2.

HPLC-MS/MS retention time and calibration data for phenolics are presented in the Appendix A. The peaks of chromatograms were identified using analytes and standards retention time compliance (Appendix A). The compounds’ quantity was calculated using linear regression correlation equations derived from the standard calibration curve (Appendix A).

### 2.6. Determination of Antioxidant Activity

ABTS^•+^ radical cation decolorization assay was applied according to the methodology described by Re et al. [29]. A volume of 3 mL of solution (absorbance 0.800 ± 0.02) was mixed with 10 μL of the tested extract. A decrease in absorbance was measured at a wavelength of 734 nm after keeping the samples in the dark for 30 min.

CUPRAC (Cupric ion reducing antioxidant activity) solution included copper (II) chloride (0.01 M in water), ammonium acetate buffer solution (0.001 M, pH = 7), and neocuproine (0.0075 M in ethanol) (ratio 1:1:1). During the evaluation, 3 mL of CUPRAC reagent was mixed with 10 µL of extracts. An increase in absorbance was recorded after 30 min at a wavelength of 450 nm [30].

The ferric-reducing antioxidant power (FRAP) assay was carried out as described by Benzie and Strain [31]. The working FRAP solution included 2,4,6-tri-2-pyridinyl-1,3,5-triazine (TPTZ) (0.01 M dissolved in 0.04 M HCl), FeCl_3_×6H_2_O (0.02 M in water), and acetate buffer (0.3 M, pH 3.6) at the ratio of 1:1:10. A volume of 3 mL of a freshly prepared FRAP reagent was mixed with 10 μL of the tested extract. An increase in absorbance was recorded after 30 min at a wavelength of 593 nm.

The antioxidant activity of tested extracts was calculated from the Trolox calibration curve and expressed as μmol Trolox equivalent (TE) per gram. TE was calculated according to the following formula:TE = (C × V)/m (μmol/g)

C: TE concentration of Trolox established from the calibration curve (in μM); V: the volume of the extract (in L); m: the weight of herbal material (in g).

### 2.7. Statistical Analysis

The experiments were performed in triplicate. The mean values and standard deviations were calculated using MS Excel (Microsoft, Redmond, WA, USA). One-way analysis of variance (ANOVA) and the post hoc Duncan test were employed for statistical analysis using IBM SPSS Statistics 29 software. The differences were considered statistically significant at the *p* < 0.05 level. Pearson’s linear correlation was applied to measure the statistical relationship between CUPRAC, ABTS^•+^, DPPH parameters, TPC, TFC, proanthocyanidins, and hydroxycinnamic acid derivative content.

## 3. Results and Discussion

### 3.1. Determination of Phenolic Compounds

In our study, the total amount of phenolic compounds (TPC) in the berries of *A. kolomikta* and *A. arguta* cultivars ranged from 134.45 ± 36.45 to 227.46 ± 1.99 mg GAE/g DW and from 35.67 ± 3.64 to 106.98 ± 11.92 mg GAE/g DW, respectively (Figure 2a). *A. kolomikta* cultivar ‘Aromatnaja’ accumulated the most TPC, while ‘Izumrudna’ was distinguished for the largest amounts of TPC among *A. arguta* cultivars. In our previous study, leaves and berries of *A. kolomikta* cultivar ‘Aromatnaja’ also showed high levels of bioactive compounds [32]. Studies of other *A. arguta* cultivars also confirmed high amounts of TPC in the berries (from 2443.3 mg/100 g DW to 6679.18 mg/100g DW) [10], which is consistent with the results of our research. However, compared to the cultivars of *A. deliciosa* species, the berries of *A. kolomikta* and *A. arguta* were several times richer in TPC [33]. Currently, phenolic compounds are of greater interest because of their biological activities and effect on human wellness and prevention of diseases.

The amount of TPC composes approximately 35% of all secondary metabolites found in *Actinidia* spp. It is important to note that in various plant parts of different *Actinidia* species, as many as 287 phenolic compounds, including 95 simple phenols and 192 polyphenols, have been found [18]. Comparing both studied species, similar trends were determined in terms of the total amounts of proanthocyanidins since considerable variation was found (Figure 2b). The average proanthocyanidins content expressed as epicatechin equivalent (mg EE/g DW) ranged from 439.97 ± 18.65 mg EE/g DW (*A. arguta*) to 1439.31 ± 25.11 mg EE/g DW (*A. kolomikta*). Thus, the amounts of proanthocyanidins were more than three times higher in berries of *A. kolomikta*. As Qi et al. [34] have confirmed, proanthocyanidins are safe and efficient natural antioxidants with potential application value. The effect of proanthocyanidins on specific diseases is proven. Conducted studies suggested that the bioactive compounds present in *A. arguta* fruit have the potential to impact glioblastoma growth by reducing cancer self-renewal [35]. As Dixon et al. emphasized, proanthocyanidins give flavor and astringency to beverages such as fruit juices and teas and are recognized as having beneficial effects on health. These compounds are also important for the growth and development of the plants themselves, as it has been established that their major function is to protect against diseases, pests, and herbivores [36].

When comparing both species and their cultivars according to the amounts of total flavonoids (TFC) and hydroxycinnamic acid derivatives, high intraspecific and interspecific variability was determined (Figure 2c,d). The total amount of hydroxycinnamic acid derivatives for *A. kolomikta* cultivars ranged from 3.96 ± 0.18 mg CAE/g DW to 89.00 ± 10.34 mg CAE/g DW (chlorogenic acid/g dry weight), and, for *A. arguta* cultivars, from 4.08 ± 0.70 mg CAE/g DW to 7.24 mg CAE/g DW. The TFC amounts expressed in mg RE/g DW also varied in the tested extracts depending on the cultivar. In berries of *A. kolomikta* and *A. arguta*, the average TFC contents were determined to be 1.58 ± 0.23 mg RE/g DW and 0.615 ± 0.08 mg RE/g DW (rutin equivalent/g dry weight), respectively (Figure 2c). These results confirmed that the most significant average amounts of all classes of phenolic compounds studied were characteristic of *A. kolomikta* cultivars. Berries of the cultivar ‘Aromatnaja’ accumulated significantly more hydroxycinnamic acid derivatives and were characterized by large amounts of proanthocyanidins and total amounts of flavonoids (TFC).

In summary, berries containing these compounds are promising for wider cultivation and application due to their exceptional effects on human health. *A. arguta* and *A. kolomikta* are both highly resistant to pests and diseases [23]. The role of phenolic compounds in horticultural plants in their resistance to fungal and bacterial diseases or pests is of fundamental and practical interest [32,36].

### 3.2. Quantitative Composition of Phenolic Compounds

In this study, phenolic compounds in berry extracts of cultivars of *A. kolomikta* and *A. arguta* were identified and quantified using UHPLC-ESI-MS/MS analysis. As shown in Table 3, 19 phenolic compounds (4 phenolic acids, 8 flavonols, 2 flavones, and 5 flavon-3-ols) were identified.

Four phenolic acids and hydroxycinnamic acid derivatives (neochlorogenic, chlorogenic, ferulic, and caffeic) were found in *A. kolomikta* and *A. arguta cultivars*. Chlorogenic acid was notably predominant, and the largest amount of chlorogenic acid (211.74 ± 6.24 µg/g DW) was found in berries of the *A. kolomikta* cultivar ‘Aromatnaja’. The lowest chlorogenic acid content (4.22 ± 0.58 µg/g DW) was determined in berries of the *A. arguta* cultivar ‘Purpurova Sadova’. According to other authors, *A. arguta* berries accumulated the most neochlorogenic acid, but another cultivar (‘Geneva’) was studied [37]. Macedo et al. [35] have presented investigations whose results supported lower amounts of both chlorogenic and neochlorogenic acids compared to our results: 0.217 µg/mg DW and 0.0528 µg/mg DW, respectively. In berries of both *A. kolomikta* and *A. arguta* cultivars, ferulic acid was found in smaller amounts compared to other hydoxycinnamic acid derivatives. However, the literature indicates that ferulic acid has especially strong antioxidant activity and is much less affected by pH changes than other phenolic acids, such as chlorogenic, caffeic, and gallic acids [38]. All the more so, as the studies of both types of berries confirmed a strong protective effect of ferulic acid (in vitro) against advanced glycation end-products (AGEs) formation with Pearson’s correlation coefficient of 0.9982 [12]. This confirms the health-promoting potential of the studied berries.

The total content of flavonols in berry extracts of *A. kolomikta* and *A. arguta* cultivars ranged from 2.64 µg/g DW to 1078.48 µg/g DW. The most common flavonols identified in all berry extracts were kaempferol-3-*O*-glucoside, isoquercitrin, and hyperoside. Evaluating the variety and amounts of flavonols, our results showed that rutin was, on average higher in *A. arguta* berries, except for the cultivar ‘Izumrudna’; however, *A. kolomikta* cultivars differed in that they had significantly higher amounts of kaempferol-3-*O*-glucoside. The largest amount of isoquercitrin (1078.48 ± 27.35 µg/g DW) was found in berry samples of the *A. kolomikta* cultivar ‘Aromatnaja’. Among the cultivars of *A. arguta*, ‘Kijivskaja Krupnoplidna’ was characterized by a higher amount of isoquercitrin—621.82 ± 7.87 µg/g DW. The antidiabetic activity of both isoquercitrin and hyperoside isolated from *A. arguta* fruits was confirmed by Kurakane et al. [39]. Furthermore, phloridzin was found in all the extracts examined. In our previous studies, this compound was identified only in the leaves of one cultivar [32].

Two compounds of the flavones group were identified in the berry extracts luteolin-4-*O*-glucoside and luteolin-7-rutinoside. The amounts of luteolin-4-*O*-glucoside varied from 8.81 ± 1.74 µg/g DW to 80.97 ± 6.14 µg/g DW in berries of *A. kolomikta*. Meanwhile, berries of *A. arguta* cultivars accumulated from 6.95 ± 0.48 µg/g DW to 14.38 ± 2.27 µg/g DW. The most luteolin-7-rutinoside (87.40 ± 6.58 DW µg/g) was found in berry samples of the cultivar ‘Aromatnaja’. These flavones are also found in *Actinidia deliciosa* berries [40]. According to our available data, these compounds have been detected in *A. kolomikta* berries for the first time, and in some cultivars, their amounts are significantly higher than in the berries of other *Actinidia* species [33,41]. Studies over the last two decades have revealed the therapeutic potential of these compounds to reduce Alzheimer’s disease symptoms in both in vitro and in vivo models [42]. The ability of luteolin to inhibit angiogenesis, induce apoptosis, and prevent carcinogenesis was also substantiated [43].

From the group of flavan-3-ols, (−)-epicatechin, (+)-catechin, procyanidin B1, procyanidin B2, and procyanidin C1 were found. The total content of flavon-3-ols in the berry extracts ranged from 10.41 µg/g to 465.27 µg/g DW. Since the total content of flavan-3-ols in the berries of *A. kolomikta* cultivars significantly exceeded the amounts determined in *A. arguta*, the same trend was also evident when evaluating the amounts of specific flavan-3-ols among different cultivars (Table 3). *A. kolomikta* berries accumulated exceptionally high amounts of procyanidin B2, which ranged from 207.20 ± 8.20 µg/g DW to 465.27 ± 32.68 µg/g DW. Previously, Zhang et al. reported that *A. arguta* berries contain (−)-epicatechin and (+)-catechin, which has shown significant anti-inflammatory activity [44]. Other authors emphasized the special biological potential of proanthocyanidins and their importance in performing neuroprotective, cardioprotective, lipid-lowering, anti-obesity, and other important functions in the human organism Qi et al. [34].

The results of the qualitative composition of phenolic compounds obtained in the previous research also showed the presence of flavan-3-ols, phenolic acids, and flavonols in *A. kolomikta* berries [32]. According to the present study, ferulic acid, luteolin-*O*-glucoside, luteolin-7-rutinoside, procyanidin B1, and procyanidin B2 were found for the first time in *A. kolomikta* berries. Since the cultivation conditions and research methods were the same, the quantitative and qualitative differences could be because the plant material was collected in different years. Climatic conditions such as drought stress and temperature are important environmental factors that restrict plant growth and alter tissue chemical composition and may significantly influence the synthesis of biologically active substances in plants [45]. In this study, both the intraspecific and interspecific diversity of phenolic compounds was investigated, identifying the most nutritionally important compounds. In summary, berries of winter-resistant *A. kolomikta* and *A. arguta* species contain significant levels of biologically active substances that have physiological and biochemical benefits and are important for human health.

### 3.3. Determination of Antioxidant Activity

*A. kolomikta* cultivars predominantly showed stronger antioxidant activity as measured by the CUPRAC method, while the antioxidant activity of the berry extract of the cultivar ‘Sentiabrskaja’ (206.00 ± 60.56 µmol TE/g DW) was characterized as very close to that of *A. arguta* cultivar ‘Izumrudna’ (206.80 ± 2.58 µmol TE/g DW) (Figure 3).

A similar situation was found when evaluating the variation in the antioxidant activity using the FRAP method. The values of antioxidant activity in berry extracts of *A. kolomikta* and *A. arguta* reached from 411.26 ± 25.24 µmol TE/g DW to 899.27 ± 1.25 µmol TE/g DW and from 60.26 ± 0.49 µmol TE/g DW to 558.70 ± 10.55 µmol TE/g DW, respectively. In conclusion, it can be stated that the berries’ antioxidant activity of both *Actinidia* species varied significantly among different cultivars of the same species, although the majority of *A. kolomikta* cultivars were characterized by higher antioxidant activity values using CUPRAC and FRAP as well as ABTS^•+^ methods.

The relationship between TPC content and antioxidant activity values is presented in Figure 4a–c. The lower amounts of TPC in the berries of *A. arguta* cultivars also resulted in lower antioxidant activity, as identified using all three methods. Moreover, even higher amounts of TPC in berries of the ‘Izumrudna’ cultivar did not significantly influence the antioxidant activity compared to *A. kolomikta* cultivars. This could be explained by the effect of another strong antioxidant-ascorbic acid because the concentration of this compound in the berries of *A. kolomikta* was determined to be several times higher than that of *A. arguta* [46,47]. Different phytochemicals contribute to the total antioxidant capacity in different proportions. Studies on different strawberry cultivars have shown that 30% of the contribution was due to vitamin C [48].

Pearson’s correlation coefficient was applied to evaluate the relationship between the antioxidant activity and secondary metabolite contents, as shown in Figure 5.

The data on the *A. arguta* species showed a significant positive correlation between TPC and TFC, proanthocyanidins, hydroxycinnamic acid derivatives, FRAP, and CUPRAC, as well as between TFC and proanthocyanidins and FRAP (*p* < 0.05) (Figure 5). On the other hand, the data on the *A. kolomikta* species showed a positive correlation between TPC and TFC, proanthocyanidins, hydroxycinnamic acid derivatives, and CUPRAC, while a high positive correlation between TFC versus proanthocyanidins, hydroxycinnamic acid derivatives, ABTS^•+^, and CUPRAC was found. Other authors have reported that the correlation analysis of various *Actinidia* genotypes revealed a strong positive dependence between antioxidant activity measured using the DPPH method versus vitamin C contained in fruit and TPC [49].

## 4. Conclusions

This study presents the results of investigations on the content of the important biologically active constituents of TPC in berries of frost-resistant *A. kolomikta* and *A. arguta* species and cultivars, along with their antioxidant activity. Comparing the amounts of TPC, it was found that *A. kolomikta* and *A. arguta* berries accumulated, on average, 177.80 mg GAE/g DW and 54.45 mg GAE/g DW, respectively. The amounts of total flavonoids, hydroxycinnamic acid derivatives, and proanthocyanidins in berries of *A. kolomikta* cultivars were also statistically reliably higher. Flavan-3-ols, flavones, hydroxycinnamic acids, and flavonols were determined by the analysis of the qualitative and quantitative content of phenolic compounds using the UHPLC-ESI-MS/MS technique. Our data indicated that berries of both hardy kiwi species were major sources of natural antioxidants. The correlation analysis showed that the total phenolic and total flavonoid compounds were the main contributors to antioxidant activity. Variation in antioxidant activity determined by CUPRAC, FRAP, and ABTS^•+^ methods was strongly influenced by different cultivar characteristics. In summary, the identified phenolic compounds are promising for more comprehensive research and application due to their exceptional effects on human health. Further investigations are necessary to broadly assess the properties of secondary metabolites and their influence on plant resistance to fungal and bacterial diseases and pests.

## Figures and Tables

**Figure 1 antioxidants-13-00372-f001:**
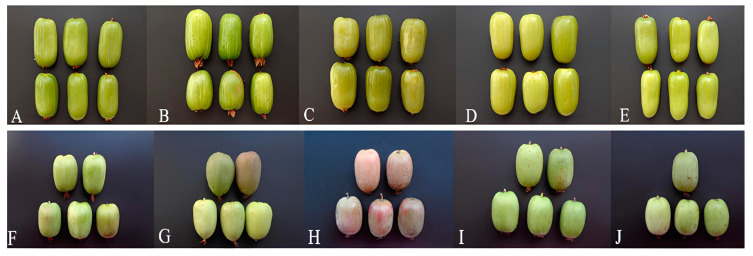
Morphological variability of *Actinidia kolomikta* (Maxim.) (**A**–**E**) and *Actinidia arguta* (Miq.) (**F**–**J**) cultivars. Cultivars: (**A**)—‘Aromatnaja’, (**B**)—‘Milema’, (**C**)—‘Matovaja’, (**D**)—‘Sentiabrskaja’, (**E**)—‘VIR-2’, (**F**)—‘Kijivskaja Krupnoplidna’, (**G**)—‘Figurna’, (**H**)—‘Purpurova Sadova’, (**I**)—‘Izumrudna’, (**J**)—‘Kijivskaja Hibridna’.

**Figure 2 antioxidants-13-00372-f002:**
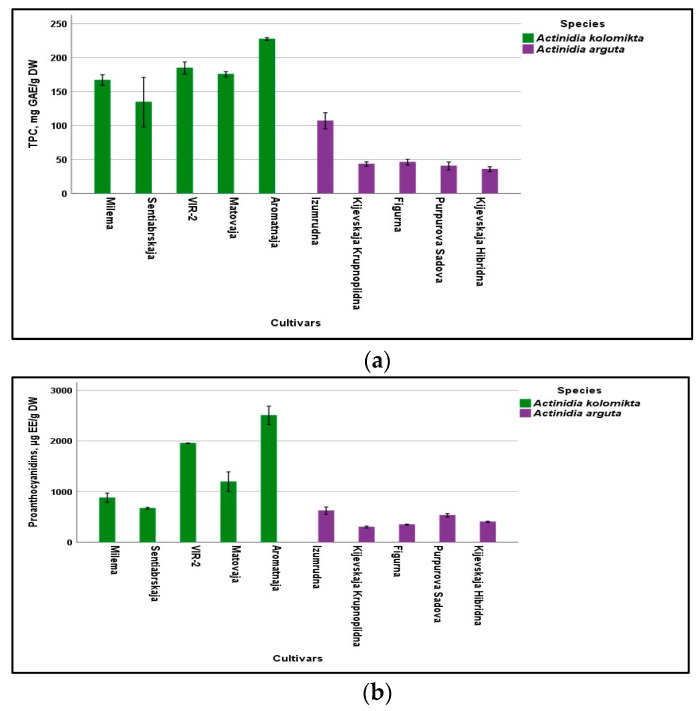
Content of total phenolics (**a**), proanthocyanidins (**b**), total flavonoids (**c**), and hydroxycinnamic acid derivatives (**d**) in berries of *A. kolomikta* and *A. arguta*. Data were expressed as mean ± standard deviation (SD) (*n* = 3). The differences were considered statistically significant at *p* < 0.05.

**Figure 3 antioxidants-13-00372-f003:**
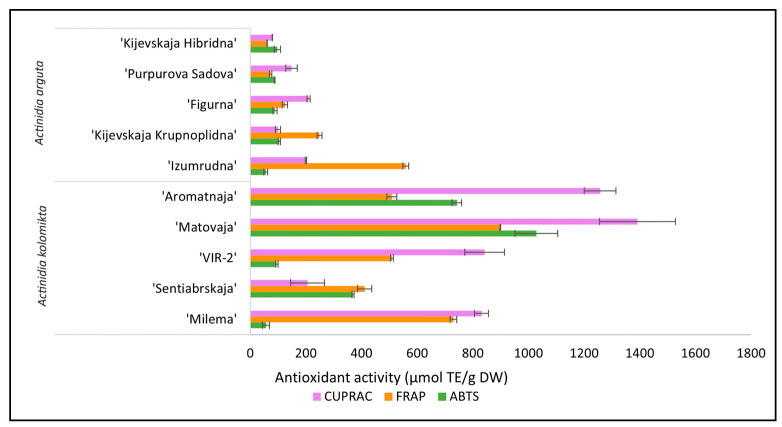
Antioxidant activity of different cultivars of *A. kolomikta* and *A. arguta* berries. Data were expressed as mean ± standard deviation (SD) (*n* = 3). Significant differences between cultivars of berries were tested, according to Duncan’s least significant difference (LSD) procedure, at the 1% significance level.

**Figure 4 antioxidants-13-00372-f004:**
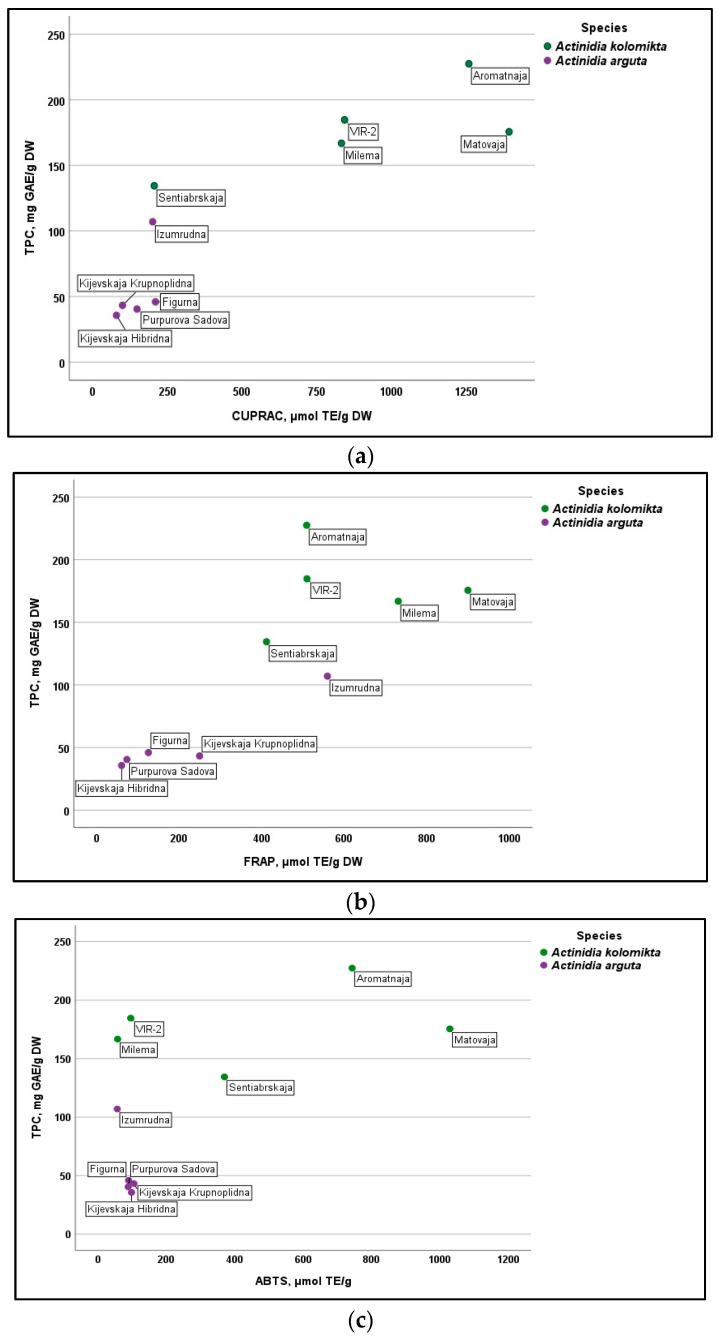
Cultivar arrangement describing (**a**) TPC (mg GAE/g DW) and CUPRAC (µmol TE/g DW) reducing activity, (**b**) TPC (mg GAE/g DW) and FRAP (µmol TE/g DW) reducing activity and (**c**) TPC (mg GAE/g DW) and ABTS (µmol TE/g DW) antiradical activity of *A. kolomikta* and *A. arguta* cultivars.

**Figure 5 antioxidants-13-00372-f005:**
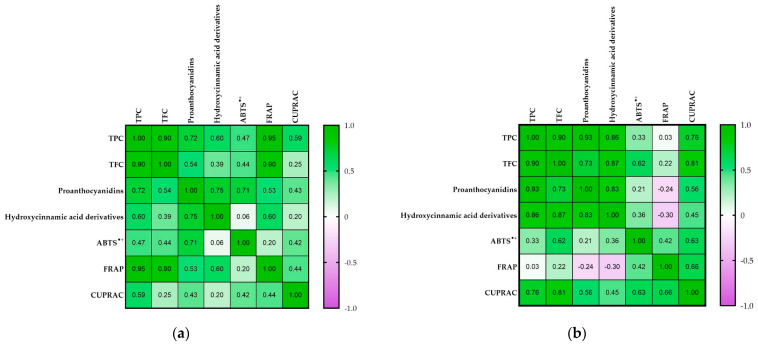
Correlation coefficients between tested bioactivities and functional constituents (**a**) *A. arguta*; (**b**) *A. kolomikta*. Pearson’s correlation was performed by using the average values of each variable (*p* < 0.05).

**Table 1 antioxidants-13-00372-t001:** Characteristics of *A. kolomikta* and *A. arguta* cultivars investigated in the present study.

Cultivar	Origin	* Average Berry Weight, g
*A. kolomikta*		
‘Sentiabrskaja’	Russia	2.04 ± 0.04 ^a^
‘Aromatnaja’	Russia	2.31 ± 0.19 ^ab^
‘Matovaja’	Russia	2.39 ± 0.17 ^b^
‘VIR-2’	Russia	2.74 ± 0.22 ^c^
‘Milema’	Lithuania	3.6 ± 0.10 ^d^
*A. arguta*		
‘Purpurova Sadova’	Ukraine	5.32 ± 0.24 ^a^
‘Izumrudna’	Ukraine	5.94 ± 0.75 ^ab^
‘Figurna’	Ukraine	6.23 ± 0.43 ^b^
‘Kijivskaja Hibridna’	Ukraine	7.18 ± 0.08 ^c^
‘Kijivskaja Krupnoplidna’	Ukraine	10.37 ± 0.39 ^d^

* Different letters denote statistically significant differences between means within the column for cultivars of each species separately (ANOVA using Duncan’s test, *p* ≤ 0.05).

**Table 2 antioxidants-13-00372-t002:** Mass spectrometry parameters for the analysis of phenolic compounds.

Compound	Parent Ion (m/z)	Daughter Ion (m/z)	Cone Voltage, V	Collision Energy, eV
Neochlorogenic acid	353	191	32	14
Kaempferol-3-*O*-rutinoside (Nictoflorin)	593	285	36	20
Quercetin 3-*O*-glucoside (Isoquercitrin)	463	301	52	28
Luteolin-4-*O*-glucoside (Juncein)	447	285	36	16
Luteolin-7-rutinoside (Scolymoside)	593	285	82	36
Procyanidin B1	577	289	50	20
Procyanidin C1	865.2	125	56	60
(+)-Catechin	289	123	60	34
Chlorogenic acid	353	191	32	14
Phloridzin	435	273	42	14
Quercetin	301	151	48	20
Isorhamnetin-3-*O*-rutinoside (Narcissin)	623	315	70	32
Ferulic acid	193	134	32	18
Procyanidin B2	577	289	50	20
(−)-Epicatechin	289	123	60	34
Caffeic acid	179	107	36	22
Kaempferol-3-*O*-glucoside (Astragalin)	447	284	54	28
Quercetin-3-*O*-rutinoside (Rutin)	609	300	70	38
Quercetin-3-*O*-galactoside (Hyperoside)	463	300	50	26

**Table 3 antioxidants-13-00372-t003:** Content of phenolic compounds (µg/g DW) in berries of different *Actinidia kolomikta* and *Actinidia arguta* cultivars.

Phenolic Compound, µg/g DW	*Actinidia kolomikta*					*Actinidia arguta*				
	‘Milema’	‘Sentiabrskaja’	‘VIR-2’	‘Matovaja’	‘Aromatnaja’	‘Izumrudna’	‘Kijevskaja Krupnoplidna’	‘Figurna’	‘Purpurova Sadova’	‘Kijevskaja Hibridna’
*Phenolic acids*										
Neochlorogenic acid	8.10 ± 3.49 ^g^	8.11 ± 1.80 ^g^	12.57 ± 3.48 ^e^	6.71 ± 1.85 ^h^	8.44 ± 2.05 ^f^	6.13 ± 0.19 ^i^	20.75 ± 0.94 ^b^	58.19 ± 3.36 ^a^	14.35 ± 1.1 ^d^	19.06 ± 2.02 ^c^
Chlorogenic acid	35.98 ± 2.78 ^h^	40.61 ± 2.89 ^g^	41.47 ± 5.95 ^f^	44.79 ± 4.02 ^e^	211.74 ± 6.24 ^a^	59.12 ± 1.55 ^c^	52.06 ± 5.04 ^d^	61.77 ± 3.56 ^b^	4.22 ± 0.58 ^j^	25.74 ± 4.06 ^i^
Ferulic acid	8.18 ± 0.54 ^d^	9.02 ± 0.81 ^b^	4.26 ± 0.39 ^h^	3.90 ± 1.03 ^i^	4.99 ± 2.20 ^g^	14.25 ± 1.45 ^a^	8.68 ± 0.33 ^c^	5.48 ± 1.23 ^f^	6.47 ± 1.16 ^e^	3.61 ± 0.21 ^j^
Caffeic acid	14.57 ± 2.96 ^d^	6.46 ± 0.29 ^e^	4.22 ± 0.36 ^j^	23.89 ± 2.72 ^b^	87.57 ± 9.73 ^a^	4.84 ± 0.13 g	4.60 ± 0.93 ^i^	5.16 ± 1.25 ^f^	4.78 ± 1.02 ^h^	15.93 ± 0.81 ^c^
*Flavonols*										
Kaempferol-3-*O*-rutinoside	26.42 ± 5.47 ^a^	6.93 ± 0.15 ^a^	12.00 ± 1.21 ^c^	17.34 ± 0.99 ^b^	11.13 ± 1.24 ^d^	5.24 ± 0.79 ^h^	4.65 ± 1.17 ^i^	2.64 ± 0.40 ^j^	5.89 ± 1.0 f	5.40 ± 0.47 ^g^
Isorhamnetin-3-*O*-rutinoside	7.58 ± 0.56 ^f^	7.49 ± 1.15 ^f^	8.90 ± 0.75 ^e^	14.55 ± 0.52 ^a^	11.45 ± 0.78 ^c^	6.77 ± 0.99 ^g^	3.88 ± 0.75 ^h^	2.94 ± 0.98 ^i^	9.51 ± 1.45 ^d^	13.31 ± 1.61 ^b^
Kaempferol-3-*O*-glucoside	387.53 ± 10.59 ^b^	65.37 ± 11.14 ^e^	109.69 ± 19.83 ^d^	301.79 ± 14.27 ^c^	564.94 ± 28.15 ^a^	49.06 ± 3.80 ^f^	6.90 ± 0.94 ^j^	44.68 ± 2.61 ^g^	7.95 ± 1.12 ^i^	12.47 ± 1.25 ^h^
Rutin	7.16 ± 0.51 ^h^	8.00 ± 0.72 ^g^	14.51 ± 0.98 ^f^	15.11 ± 1.33 ^e^	32.16 ± 2.18 ^d^	6.33 ± 0.31 ^i^	81.46 ± 6.86 ^a^	43.12 ± 0.75 ^b^	37.86 ± 1.76 ^c^	43.47 ± 1.87 ^b^
Isoquercitrin	283.49 ± 20.4 ^fe^	123.14 ± 19.40 ^i^	236.18 ± 34.19 ^h^	958.42 ± 31.95 ^b^	1078.48 ± 27.35 ^a^	99.60 ± 5.74 ^j^	621.82 ± 7.87 ^c^	383.23 ± 21.43 ^e^	593.15 ± 9.61 ^d^	257.02 ± 20.54 ^g^
Quercetin	11.23 ± 1.83 ^b^	3.51 ± 1.83 ^h^	5.36 ± 2.09 ^d^	4.83 ± 0.91 ^e^	4.20 ± 0.64 ^f^	4.27 ± 0.76 ^f^	7.48 ± 1.40 ^c^	3.76 ± 0.77 ^g^	4.79 ± 1.07 ^e^	15.93 ± 1.07 ^a^
Phloridzin	6.50 ± 3.70 ^d^	3.69 ± 1.15 ^e^	2.69 ± 0.86 ^f^	11.05 ± 1.08 ^a^	2.71 ± 0.51 ^f^	2.77 ± 0.24 ^f^	9.49 ± 1.26 ^b^	9.46 ± 1.44 ^b^	3.61 ± 1.20 ^e^	9.11 ± 0.63 ^c^
Hyperoside	91.49 ± 7.59 ^e^	50.06 ± 6.43 ^h^	78.53 ± 8.78 ^g^	262.53 ± 6.33 ^b^	295.83 ± 9.97 ^a^	47.08 ± 1.85 ^i^	25.58 ± 2.19 ^j^	116.95 ± 8.01 ^d^	156.59 ± 8.60 ^e^	79.37 ± 10.37 ^i^
*Flavones*										
Luteolin-4-*O*-glucoside	53.28 ± 4.09 ^b^	8.81 ± 1.74 ^g^	16.86 ± 4.03 ^d^	42.72 ± 4.03 ^c^	80.97 ± 6.14 ^a^	6.95 ± 0.48 ^i^	8.07 ± 1.61 ^h^	11.07 ± 0.99 ^f^	11.55 ± 0.96 ^f^	14.38 ± 2.27 ^e^
Luteolin-7-rutinoside	26.75 ± 0.84 ^b^	6.82 ± 0.69 ^e^	13.48 ± 2.24 ^d^	19.93 ± 2.86 ^c^	87.40 ± 6.58 ^a^	5.56 ± 0.83 ^f^	3.97 ± 0.19 ^h^	2.36 ± 0.48 ^i^	5.58 ± 1.23 ^f^	5.24 ± 0.62 ^g^
*Flavon-3-ols*										
(+)-Catechin	41.05 ± 2.20 ^b^	34.14 ± 5.67 ^c^	22.59 ± 3.15 ^f^	30.42 ± 4.79 ^d^	69.71 ± 5.63 ^a^	27.05 ± 3.07 ^e^	11.19 ± 1.84 ^bh^	10.41 ± 2.89 ^i^	14.33 ± 2.13 ^g^	11.10 ± 0.28 ^h^
(−)-Epicatechin	361.79 ± 43.14 ^c^	200.45 ± 11.24 ^e^	456.71 ± 14.00 ^a^	236.15 ± 12.65 ^d^	393.15 ± 34.47 ^b^	23.79 ± 3.16 ^g^	17.29 ± 3.05 ^j^	22.99 ± 1.45 ^h^	44.90 ± 5.27 ^f^	21.00 ± 2.55 ^i^
Procyanidin B1	199.42 ± 32.90 ^a^	122.20 ± 33.23 ^d^	157.56 ± 27.50 ^b^	144.40 ± 8.39 ^c^	198.04 ± 9.87 ^a^	144.10 ± 8.03 ^c^	29.83 ± 7.68 ^f^	20.72 ± 4.14 ^g^	35.97 ± 2.99 ^e^	28.25 ± 4.10 ^f^
Procyanidin C1	163.73 ± 5.19 ^c^	113.92 ± 2.98 ^f^	289.24 ± 27.60 ^a^	128.20 ± 5.75 ^d^	255.18 ± 7.93 ^b^	116.70 ± 1.56 ^e^	19.10 ± 1.29 ^dh^	12.39 ± 1.71 ^j^	36.90 ± 1.00 ^g^	18.44 ± 2.32 ^i^
Procyanidin B2	262.22 ± 16.58 ^c^	208.17 ± 11.48 ^d^	465.27 ± 32.68 ^a^	207.20 ± 8.20 ^d^	336.95 ± 11.67 ^b^	157.52 ± 12.54 ^e^	34.03 ± 3.57 ^g^	29.14 ± 2.41 ^i^	55.19 ± 3.59 ^f^	32.84 ± 3.66 ^h^

Values were expressed as mean ± standard deviation (SD) (*n* = 3); different superscript letters within the same row indicate statistically significant differences according to Duncan’s least significant difference (LSD) procedure at a 5% significance level.

## Data Availability

The data presented in this study are available in the article.

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
