# Peer review of "Phenolic Compounds in Berries of Winter-Resistant Actinidia arguta Miq. and Actinidia kolomikta Maxim.: Evidence of Antioxidative Activity"

_antioxidants, 2024, doi:10.3390/antiox13030372_

Round 1

Reviewer 1 Report

My comments are listed in the attached file

Author Response

Response to Reviewer 1 Comments 

The authors are grateful for the Reviewer’s report and feedback, which improves the quality of the manuscript. 

 The authors have already published a work on A. kolomikta (REF: 10.3390/plants11020147) where similar results were presented (phenolic content and antioxidant activity). They should clearly state in the current work which are the differences and improvements respect to the previous one (that has to be cited).

Response. Thanks to the reviewer for this important note. Our previous study analysed twelve different A.kolomikta cultivars, and the current study supplements previous results. We included the previous publication in the literature list [32] and cited it. We thoroughly compared and interpreted the divergent findings with our previous work about A. kolomikta (lines 207-209; 262-264; 297-299; 324-325).

Overall, the novelty of the results contained in the current work is low, especially considering the previous publication and the   high amount of literature data on A. arguta.

Response. Thank you for this valuable comment. We agree that the Results and Discussion section must be improved. We have supplemented the article with information about the differences between the studied cultivars, comparing them with data from previous studies. We added 11 references to the article, so the obtained results were analysed and commented on in more detail compared to the works of other authors. All additions are highlighted in the results and discussion section. Literature data on                       A. kolomikta studies are still relatively few. Another winter-resistant species A. arguta is studied in much more detail.  We also added correlation analysis to section 3.3 of the article. Your suggestions have prompted us to re-evaluate and enhance the clarity of our results and discussion part, and we thank you for your contribution to the refinement of our work (lines 238-245; 275-286; 297-311; 317-323; 324-337; 386-402).

 Some MAJOR ISSUES are present in the work. Mainly, they regard methodological aspects and need to be improved."

Response. Thank you.  We have clarified the section Materials and Methods (lines 162-165). The equipment used was specified (lines 136-137; line 142). We also performed a correlation analysis (Pearson’s test and commented on its results. Corrections to the Results and Discussion section have also been made.

Abstract 

Line 26: “applying ultra-high performance liquid chromatography (UHPLC) mass spectrometry…”. The sentence need correction: “applying ultra-high performance liquid chromatography (UHPLC) COUPLED TO mass spectrometry”. 

Response. The sentence was corrected (P. 1, line 31). The same correction was made in P. 5, line 156. 

Abstract is too generic. I suggest the authors to reformulate it with some more details on the findings of the study. 

Response. Thank you for your comment. The abstract was corrected (P. 1, lines 24-27). 

Introduction 

Line 46: delete repeated words. 

Response. Thank you for bringing to our attention this comment. The plant name was corrected. Actinidia deliciosa (A.Chev.) C.F.Liang & A.R.Ferguson (P. 2, lines 50-51).  

Figure 1: I suggest the authors to report the image with higher dimensions, keeping high resolution. 

Response. Thank you for the comment. Thank you for the comment. We acknowledge the importance of adhering to high-resolution figures in the manuscript. We have made the necessary improvements to ensure the quality of Figure 1.

Table 1: letters in superscript. 

Response. Thank you for your attention to detail, Letters were corrected in Table 1. (P. 3-4, lines 109-111).  

Methods 

 Paragraph 2.5: how was the identity of the target compounds confirmed? Measuring a single MS1-MS2 transition does not allow to identify a single compound since other isomers can show the same transition. Furthermore, how were these quantified? Please report all the missing details. 

Response. Thank you for your insightful comment. The peaks of chromatograms were identified using analytes and standards retention time compliance. The compounds' quantity was calculated using linear regression correlation equations derived from the standards calibration curve (P. 5, lines 165).

Results 

Figures 2 needs to be reported with right proportions. 

Response. Thank you for bringing this up to us. Figure 2 was corrected. 

Paragraph 3.3.: actually, results of correlation analysis are not presented here. The authors merely compared phenolic content to antioxidant effects of the tested samples. I suggest to perform a Pearson’s or Spearman’s test (depending on data distribution) to verify correlations. 

Response. Thank you for this comment. Thank you for this comment. Following your suggestion, we incorporated correlation analysis, which significantly improves the readability and comprehensibility of our research. Pearson’s linear correlation was applied to measure the statistical relationship between CUPRAC, ABTS•+, DPPH parameters, TPC, TFC, proanthocyanidins, and hydroxycinnamic acid derivatives content (Fig. 5; lines 395-402).

We are truly grateful for your expertise and experience in providing valuable feedback on our review manuscript. Your guidance upmost enhances the quality of our work.

Sincerely, 

Reviewer 2 Report

 The manuscript entitled “Phenolic compounds in berries of Actinidia arguta Miq. and Actinidia kolomikta Maxim.: evidence of antioxidative activity” by Česonienė et al. intends to evaluate the antioxidant properties of A. arguta and A. kolomikta extracts, using as solvent ethanol. The authors applied TPC, TFC, proanthocyanidin content, CUPRAC, FRAP and ABTS methods to evaluate the extracts from five different cultivars, as well as UHPLC to identify the phenolic compounds. The manuscript is in the scope of the Antioxidants. The paper is well written, however the results and discussion part is poor. 

Other studies such as anti-diabetic, anti-fungal, antimicrobial or cytotoxic assays (or others) should be performed in order to improve manuscript and differentiate from other manuscripts published.

In section 2.3, the authors should specify the equipment used in extraction method.

Some abbreviations as DMCA and CUPRAC should be introduced.

The authors should introduce abbreviations as TFC in Materials and Methods section, as well as the expression of results of proanthocyanidins content.

In Results and Discussion Section, the authors should rewrite and improve. Specifically, in 3.1 and 3.3 parts, the results from different cultivars should be compared and discussed, justifying the differences and comparing with other studies yet published. 

In lines 192-195, the authors should put references.

Author Response

Response to Reviewer 2 Comments

The authors are grateful for the Reviewer’s report, which significantly improves the manuscript.  We sincerely appreciate your expertise and comments, which we are addressing below.

The paper is well written, however the results and discussion part is poor.

Response. Thank you for your helpful comment regarding the discussion part. We agree that the discussion can be improved in the first version of the manuscript. According to your suggestion revised version of the article was comprehensively supplemented with new literature sources.

Other studies such as anti-diabetic, anti-fungal, antimicrobial or cytotoxic assays (or others) should be performed in order to improve manuscript and differentiate from other manuscripts published.

Response. Thank you for the comment. We have taken your feedback into account and provided information about antidiabetic, anticancer, antioxidant etc. properties of identified phenolic compounds in the revised manuscript (references 37, 39, 40, 42, and 49), lines 238-241; lines 279-286; lines 393-402).

In section 2.3, the authors should specify the equipment used in the extraction method.

Response. Thank you. We really appreciate your attention to detail. The equipment used in the extraction method was specified (lines 136-137; line 142).

Some abbreviations as DMCA and CUPRAC should be introduced. The authors should introduce abbreviations as TFC in Materials and Methods section, as well as the expression of results of proanthocyanidins content.

Response. Thank you for your valuable comment. The explanations of indicated abbreviations were provided thoroughly in the manuscript. This addition will help readers better comprehend our research content, and we thank you for it (lines 146, 148, 176).

In Results and Discussion Section, the authors should rewrite and improve. Specifically, in 3.1 and 3.3 parts, the results from different cultivars should be compared and discussed, justifying the differences and comparing with other studies yet published.

Response. Thank you for this valuable comment. We agree that the Results and Discussion section must be improved. We have supplemented the article with information about the differences between the studied cultivars, comparing them with data from previous studies. Our previous study also discussed [32], lines 207-209; 260-263; 298-299; 324-326.

 We also added correlation analysis to section 3.3 of the article. Your suggestions have prompted us to re-evaluate and enhance the clarity of our results and discussion part, and we thank you for your contribution to the refinement of our work (lines 238-245; 275-286; 297-311; 317-323; 324-337; 386-402).

In lines 192-195, the authors should put references.

Response. Thank you for your diligence in maintaining the integrity of our references and content. We revised the added missing reference Waswa et al., 2024 [18], line 232.

The manuscript needs more comparisons with other studies and results, as well as justify the values obtained.

Response. Thank you for this comment. We added 11 references to the article, so the obtained results were analysed and commented on in more detail compared to the works of other authors. All additions are highlighted in the results and discussion section.

The manuscript should be rewritten and improved, as well as more scientific studies should be performed as antimicrobial/anti-fungal/anti-diabetic or cell assays.

Response. Thank you for your insightful comment. As per your suggestion, more literature sources are examined in the Results and Discussion section. For that purpose, the following articles by other authors are used for discussion: 12, 36, 37, 38, 40, 41, 43, 44, and 49, lines 241-246; 283-286; 308-311; 319-323. We appreciate your input on this topic and valuable suggestions for improvements. We agree that more scientific studies should be performed, and we hope to continue the evaluation of different beneficial properties. Together with comprehensive antioxidant profiles of investigated cultivars, we aim to promote academic results. However, the authors believe that in the current research, biochemical composition, antioxidant activities, and their correlation already introduce the readers to explicit information.

Minor editing of English language required.

Response. The English language of the article has been corrected.

Thank you again for your in-depth attention and dedication to ensuring accuracy and quality in our work.  Your comments made invaluable contributions to our research, which we are genuinely grateful for.

Sincerely,

Round 2

Reviewer 1 Report

The work has been improved, however still there are lacks in the part regarding the phytochemical characterization of the extract.

- The Authors state that: "The peaks of chromatograms were identified using analytes and standards retention time compliance." However, it is not clear if standards of each analysed compound were coinjected. There is no mention on the respective retention times, nor in the text nor in tables. No exemplificative chromatograms of a mixture of standards and the analysed extract are reported. Authors are required to improve this part of the work adding the missing information/material.

- "The compounds' quantity was calculated using linear regression correlation equations derived from the standards calibration curve (P. 5, lines 165)." Also here it is not clear if a calibration curve was made for all the standards, or if some of them were taken as references for the semi-quantification of compounds with similar structures. Authors have to add the missing information and indicate the equation of the calibration curves.

- Results: a Figure showing exemplificative chromatograms of 1) a mixture of the standard compounds used, and 2) the analysed extract has to be added.

n/a

Author Response

Major comments

The work has been improved, however still there are lacks in the part regarding the phytochemical characterization of the extract.

Dear Reviewer,

Thank you for your valuable feedback about our manuscript which improves the quality significantly. Below we are answering your comments and suggestions:

Comment 1

- The Authors state that: "The peaks of chromatograms were identified using analytes and standards retention time compliance." However, it is not clear if standards of each analysed compound were coinjected. There is no mention on the respective retention times, nor in the text nor in tables. No exemplificative chromatograms of a mixture of standards and the analysed extract are reported. Authors are required to improve this part of the work adding the missing information/material.

Answer 1

Thank you for your comment. The standards of each analysed compound were coinjected. A slight but consistent retention shift is visible in the provided data. Also, we added supplementary Table S1 which includes retention times and exemplificative chromatograms in supplementary Figures S1, S2, and S3. This information was provided in the section Materials and Methods (lines 170-171) and Supplementary Materials (lines 427-435).

Comment 2

- The compounds' quantity was calculated using linear regression correlation equations derived from the standards calibration curve (P. 5, lines 165)." Also here it is not clear if a calibration curve was made for all the standards, or if some of them were taken as references for the semi-quantification of compounds with similar structures. The authors have to add the missing information and indicate the equation of the calibration curves.

Answer 2

We genuinely value your time and patience to improve our manuscript. Thank you. For clarification, calibration curves were made for all the standards. Also, no semi-quantification was performed. Moreover, as per your valuable suggestion, equations of the calibration curves were provided in supplementary Table S1.

Comment 3

- Results: a Figure showing exemplificative chromatograms of 1) a mixture of the standard compounds used, and 2) the analysed extract has to be added.

Answer 3

Once again, thank you for your comment. Chromatograms of 3 mixtures of the standard compounds used and the analysed extract were provided in supplementary Figures S1, S2, and S3.

Sincerely,